# Colloidal gelation with non-sticky particles

**Yujie Jiang** [1,2] ✉ **& Ryohei Seto** [1,3,4] ✉

Colloidal gels are widely applied in industry due to their rheological character—no flow takes place below the yield stress. Such property enables gels to maintain uniform distribution in practical formulations; otherwise, solid components may quickly sediment without the support of gel matrix. Compared with pure gels of sticky colloids, therefore, the composites of gel and non-sticky inclusions are more commonly encountered in reality. Through numerical simulations, we investigate the gelation process in such binary composites. We find that the non-sticky particles not only confine gelation in the form of an effective volume fraction, but also introduce another lengthscale that competes with the size of growing clusters in gel. The ratio of two key lengthscales in general controls the two effects. Using different gel models, we verify such a scenario within a wide range of parameter space, suggesting a potential universality in all classes of colloidal composites.

Sticky colloidal particles diffuse and aggregate into clusters until forming a ramified, space-spanning network, i.e., colloidal gel[1,2]. Due to the load-bearing structure, colloidal gels behave as soft solids with finite yield stress beyond which flow occurs[3,4]. This rheological signature enables gels to be widely applied in industry, ranging from foodstuffs and personal care products to pharmaceutics and biotechnology[5–7]. Through experiments and simulations, particulate gels of monodisperse attractive (i.e., single-component) colloids have been extensively studied in various aspects (e.g., structure[2], dynamics[5], and rheology[8]), while theories have been proposed to approach a priori prediction (such as ref. 9). By contrast, realistic gel-like materials, which usually contain multiple components, remain less probed. This is partially due to the lack of proper model systems in experiments, while the intrinsic complication of polydispersity also limits the progress in fundamental understanding. While recent attention on composite systems increases[10–15], most studies still remain on the phenomenological level of ac hoc models.

Among the diverse array of multi-component systems, the combination of gel matrix and solid fillers is prototypical in practical applications. For example, polymeric nanocomposites have remarkable mechanical properties and are ubiquitous in sensors, civil engineering, and microbial applications[16]. Progress has been made in understanding such composites, which, in the absence of strong filler–matrix interactions, can be described by conventional continuum mechanics[17]. Because of a large gap between the constituent sizes, empirical approaches (such as Krieger–Dougherty law) have been found to describe the basic behavior by assuming a continuum background[18,19].

However, the continuum assumption no longer holds if the characteristic sizes of each component are comparable[20]. This is the case when replacing the polymer matrix in polymeric nanocomposites with a colloidal gel network, where the typical lengthscales are all micron-sized. The interplay between these lengthscales generates novelty. Though not yet fully understood, such biphasic mixtures receive increasing attention[21,22], and recent work reports a unique flow-switched bistability[23], which has never been observed in regular gels. These observations suggest the essential role of filler (or inclusion[17]) particles. While researchers have recently focused on the gelled state of these composites[17,24], the inclusion effect on gelation dynamics is still unclear.

Using numerical simulations, we aim to shed light on the understanding of colloidal gelation with non-sticky particle fillers. The system we investigate is composed of sticky colloids, which can form a percolating gel network on their own, and non-sticky (NS) particles, which are hard spheres. As the latter stick neither to the gel colloids nor to themselves, they do not participate in gelation directly. Then the intuitive assumption seems to view NS particles as confinement to the gel part by compressing the available volume. Through extensive

[1]Wenzhou Key Laboratory of Biomaterials and Engineering, Wenzhou Institute, University of Chinese Academy of Sciences, 325000 Wenzhou, Zhejiang, China. [2]School of Physical Sciences, University of Chinese Academy of Sciences, 100049 Beijing, China. [3]Oujiang Laboratory (Zhejiang Lab for Regenerative Medicine, Vision and Brain Health), 325000 Wenzhou, Zhejiang, China. [4]Graduate School of Information Science, University of Hyogo, 650-0047 Kobe, Hyogo, Japan. ✉e-mail: jiangyujie@ucas.ac.cn; seto@ucas.ac.cn

exploration of the parameter space, we show that the interplay between comparable lengthscales also plays a vital role and, in some cases, dominates over the confining effect and greatly impedes the gelation process. The ratio of two key lengthscales, i.e., the characteristic size of gel and the NS-particle spacing, offers a robust measure for such interplay, which controls the cluster growth during gelation. We verify this scenario over a wide range of compositions and particle sizes in different types of colloidal gel.

## RESULTS

### Colloidal gelation

A variety of attractions lead to colloidal gelation in nature, such as van der Waals forces, depletion forces, and hydrophobic interactions[25]. In this work, we investigate gelation under strong attractions ($U_{att} \gg k_B T$) within a wide range of volume fractions ($0.03 \leq \phi \leq 0.3$). We consider two representative contact models. The first model refers to typical attractions which drive particles to aggregate with only radial forces, as shown in Fig. 1a (blue). Such conservative attractions are characterized by pairwise potentials and mostly apply to smooth particles[2] without tangential constraints, i.e., particles in contact can freely slide and rotate. By contrast, the second model constrains tangential pairwise motions (sliding, rolling, and twisting) with the presence of attraction, Fig. 1a (red). As suggested in ref. 26, we term the first contact as attraction (att) and the second model with tangential constraints as adhesion (adh).

The difference in contact interactions leads to different micromechanics, such as bending rigidity[27] and isostaticity[28]. Based on the Maxwell criteria[29], mechanical stability for frictionless particles in 3D requires an average contact number $N \geq N_c = 6$. Following[30], we consider the isostaticity condition as microstructural information in attractive and adhesive systems and determine gelation accordingly. In particular, we extract particles with $N \geq N_c$ and check their connectivity (see the "Methods" section), Fig. 1b. The gelation point determined by this method has been verified, in both experiments[31] and simulations[24], to agree well with that from macroscopic rheology. Therefore, we apply this gelation criterion throughout this work, with $N_c^{att} = 6$ for attractive gels and $N_c^{adh} = 2$ for adhesive gels[28].

Our simulations start from a random, homogeneous configuration and evolve following Langevin dynamics for up to $10^4$ times of Brownian time $\tau_B \equiv \pi \eta d^3 / 2k_B T$ (where $\eta$ refers to fluid viscosity and $d$ to colloid diameter). More simulation details can be found in the Methods section. For both gels, the time required for gelation $t_g$ decreases with the volume fraction $\phi$ in a power-law manner, Fig. 1c. Fitting of attractive gel data (blue) gives an exponent of −3.7, while that

of adhesive gels (red) exhibits a lower exponent −2.1. Detailed fitting results are as follows:

$$t_g^{att} \approx 0.021 \times \phi^{-3.7}, \tag{1}$$

$$t_g^{adh} \approx 0.011 \times \phi^{-2.1}. \tag{2}$$

At the same volume fraction $\phi$, it takes adhesive colloids less time to gel than the attractive ones. The exponents roughly agree with the values in other literature[32–34], justifying our gel simulations as well as the gelation criterion we used.

To capture the structural evolution during gelation, we measure the static structure factor $S(q)$ for both attractive ($\phi = 0.1$) and adhesive ($\phi = 0.05$) gels, Fig. 1d. $S(q)$ is originally flat due to the homogeneous randomization for the initial configuration. As time increases (indicated by arrows in Fig. 1d), a peak at intermediate wavenumber $q$ appears, grows, and shifts to a lower $q$. Thus, a characteristic lengthscale $\xi \equiv 2\pi/q_0$ (where $q_0$ refers to the peak wavenumber) increases during gelation, plausibly representing the cluster growth. Note that the low-$q$ peak is absent at $\phi = 0.5$ (Supplementary Note 1), indicating a homogeneous attractive glass (AG) state. The gel-to-glass transition explains the slight deviation from the power-law scaling of $t_g$, Fig. 1c (blue).

While the two systems exhibit similar $S(q)$ evolutions, the fractal dimensions $d_f$ inside clusters, which can be estimated from the slope of $S(q) \sim q^{-d_f}$, are different[35]. According to Fig. 1d (top), the clusters in attractive gel are rather compact ($d_f \approx 3$) at short range, suggesting gelation via the typical arrested-phase-separation route[2,25]. As $q$ decreases, the fractal dimension $d_f$ drops to 2 (consistent with the value reported in ref. 7), indicating a relatively open structure at larger lengthscales. We attribute this to the emergent bending rigidity of bigger building blocks, e.g., tetrahedrons composed of attractive particles.

By contrast, the gel network in the adhesive gel is more open and ramified with a lower fractal dimension $d_f \approx 1.8$, as expected by diffusion-limited cluster–cluster aggregation (DLCA)[25]. Since the clusters in adhesive gels are looser than those in attractive gels, it is easier for adhesive colloids to percolate at the same volume fraction $\phi$, i.e., lower gelation time $t_g$. The above results show that the two models we used lead to two different types of colloidal gels.

### Gelation with NS particles

In the presence of NS particles, sticky colloids can still diffuse and aggregate into a percolating gel network, such as Fig. 2a. Analogous to

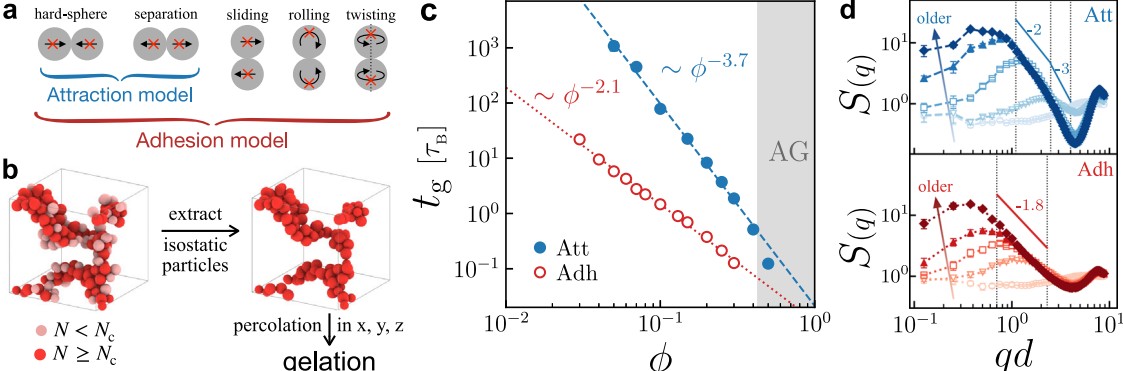

**Fig. 1 | Gelation dynamics of colloidal gels. a** Sketches of possible pairwise motions and two contact models with constraints shown in red crosses. **b** Schematic gelation determination. See details in the Methods section. **c** Gelation times $t_g$ as functions of volume fraction $\phi$ in different gels. The blue dashed line and red dotted line are power-law fittings with results shown in Eqs. (1) and (2). The gray

region denotes the attractive glass (AG) regime. **d** Evolution of structure factors $S(q)$ in an attractive gel ($\phi = 0.1$, top) and an adhesive gel ($\phi = 0.05$, bottom). The open and filled symbols represent $S(q)$ of snapshots before and upon gelation, respectively, and the arrows indicate time evolutions (from bottom to top: $t/\tau_B = 0$, 1, 10, 100, and 1000). Solid lines indicate the slope at intermediate wavenumbers.

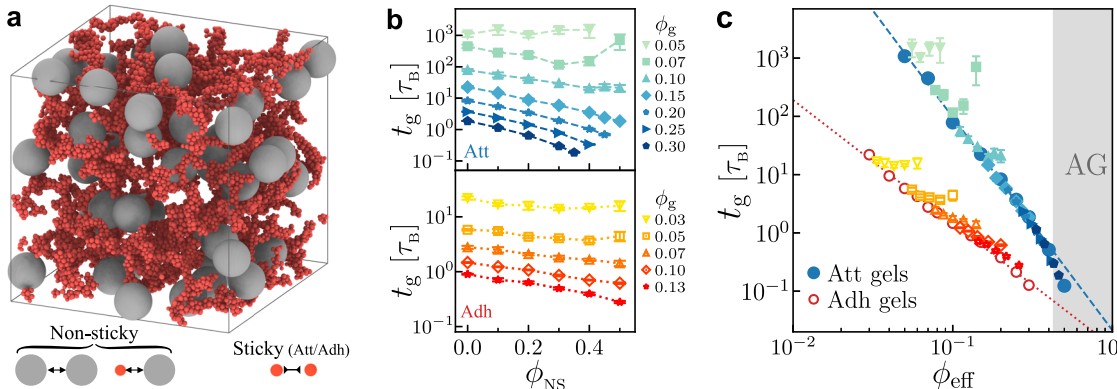

**Fig. 2 | Colloidal gelation with non-sticky particles. a** 3D rendering of a gelled attractive system of $d_{NS} = 8d$ at $\phi_g = 0.05$ and $\phi_{NS} = 0.1$. Red and gray spheres represent sticky colloids and non-sticky particles, respectively. **b** Gelation times $t_g$ of attractive (top) and adhesive (bottom) systems of $d_{NS} = 8d$ vary as functions of $\phi_{NS}$ at different $\phi_g$. Data with $\phi_{NS} = 0$ refer to colloidal gels. **c** Plot of the same data in **b** with $t_g$ versus $\phi_{eff}$ (defined in Eq. (3)). The dashed blue line and dotted red line are the power-law fittings of gel data in Fig. 1b, also see Eqs. (1) and (2).

colloidal gels where $\phi$ directly controls gelation, the gelation in binary systems depends on the volume fractions of gel colloids $\phi_g$ and NS particles $\phi_{NS}$. Since NS particles do not directly participate in the formation of gel network, we expect them to geometrically confine gelation and compress the free volume $V_{free}$ for sticky colloids. The reduction in $V_{free}$ then leads to an effective increase in the colloidal volume fraction. If simply consider $V_{free}$ by subtracting the NS particle volume $V_{NS}$ from the total volume $V_{tot}$, we can then define an effective volume fraction $\phi_{eff}$ for gel colloids as below:

$$\phi_{eff} \equiv \frac{V_g}{V_{free}} = \frac{V_g}{V_{tot} - V_{NS}} = \frac{\phi_g}{1 - \phi_{NS}}. \quad (3)$$

Though the definition above neglects the exclusion shell around NS particles[36], it has been verified to well capture the gelation diagram with varying attractions[24]. This is probably because as clusters grow and become increasingly large and porous, the cluster–NS interpenetration[37] invalidates the application of exclusion shell.

We first focus on the case of large NS particles with $d_{NS} = 8d$, where $d_{NS}$ and $d$ refer to the sizes of NS particles and gel colloids, respectively. According to Fig. 1b and Eq. (3), the addition of NS particles is expected to decrease the gelation time $t_g$. For both models at high $\phi_g$ ($\phi_g \geq 0.15$ for attractive systems and $\phi_g \geq 0.07$ for adhesive systems), $t_g$ decreases with the volume fraction of NS-particles $\phi_{NS}$ monotonically, Fig. 2b. Plotting $t_g$ versus $\phi_{eff}$ collapses these high-$\phi_g$ data on a master curve, Fig. 2c, which converges with the gel data we have shown in Fig. 1c. In this way, regardless of contact models, the effective volume fraction $\phi_{eff}$ seems to well characterize the gelation time $t_g$ of sticky-NS composites.

At low $\phi_g$, nevertheless, the decrease in $t_g$ becomes progressively slow as $\phi_{NS}$ increases. In particular, for attractive systems at $\phi_g = 0.07$, the gelation time $t_g$ even increases with the addition of NS particles at high $\phi_{NS}$ and becomes higher than that of the pure gel, Fig. 2b (top). This conflicts with our expectation of increasing $\phi_{eff}$. Furthermore, plotting $t_g$ versus $\phi_{eff}$ shows deviation from the master curve, Fig. 2c. While the data collapse at high $\phi_g$ justifies the definition of $\phi_{eff}$, this inconsistency indicates another physics, manifesting at low $\phi_g$, that delays the gelation with NS particles.

### Lengthscale interplay and diagrams

As previously mentioned, one notable feature of gel–particle composites is the comparable lengthscales. Here we identify two key lengthscales from each constituent and attribute the abnormal deviation from $\phi_{eff}$ prediction (Fig. 2c) to their interplay. The first lengthscale is the characteristic size $\xi$ of the gel structure derived from

structure factor $S(q)$, which evolves over time (Fig. 1c). Since our focus is the gelation time $t_g$, we measure the structure factor $S(q)$ at $t = t_g$ (Supplementary Note 2) and extract a time-independent lengthscale $\xi_g \equiv \xi(t_g)$. Such lengthscale in general represents the correlation length at gelation point, Fig. 3a (inset).

For both attractive and adhesive gels, $\xi_g$ decreases with volume fraction $\phi$, Fig. 3a. Namely, the more concentrated a system is, the smaller clusters it requires to assemble into a percolating network. This is consistent with the gelation at the DLCA limit[38]. Power-law fittings on the two sets of data give similar exponents, while the lengthscale in adhesive gels $\xi_g^{adh}$ is slightly lower than that in attractive gels $\xi_g^{att}$ at the same $\phi$. Fitting results are as below:

$$\xi_g^{att} \approx 1.01 \times \phi^{-0.86}, \quad (4)$$

$$\xi_g^{adh} \approx 0.65 \times \phi^{-0.90}. \quad (5)$$

Note that the above $\xi_g$ refers to the lengthscale in pure colloidal gels and $\phi$ to the colloid volume fraction. In binary systems, the large NS particles can easily distort the large-scale structure so that the colloid–colloid structure factor $S(q)$ barely exhibits a resolvable peak at low $q$ (Supplementary Fig. 3a). We, therefore, assume that the $\phi_{eff}$ scenario also applies to the $\xi_g$ scaling, by simply replacing $\phi$ with $\phi_{eff}$ in Eqs. (4) and (5). That is, we use $\xi_g$ in an equivalent pure gel as a proxy for that in a binary composite. By comparing the void distribution[39] in pure gels and composites, we justify such assumption in Supplementary Note 3.

Apart from geometric confinement, NS particles also generate a flexible porous medium[40], in which colloids diffuse and aggregate into a gel network. Such medium is in general characterized by the pore size[20]. Here we use the spacing between NS particles $\delta$ to represent the porosity, Fig. 3b (inset). In particular, we simulate a collection of only NS particles at different volume fractions $\phi_{NS}$ and measure the average interstice $\delta$ from Voronoi cell volume (the Methods section). In the unit of $d_{NS}$, the average spacing $\delta$, diverging at $\phi_{NS} = 0$, decreases with $\phi_{NS}$ as shown in Fig. 3b. Though we measure $\delta$ in pure NS-particles systems, such quantity in binary mixtures remains almost unchanged at the same $\phi_{NS}$ (see gray and black scatters in Fig. 3b). Moreover, as gelation proceeds, $\delta$ barely varies over time (Supplementary Fig. 5). Thus, $\delta$ is time-independent and scales with NS particles' absolute, rather than relative, volume fraction.

Given other parameters in a binary system fixed, both $\xi_g$ and $\delta$ can be a priori determined by $\phi_{eff}$ and $\phi_{NS}$, respectively. Then their ratio $\gamma \equiv \xi_g/\delta$ varies as a function of $\phi_g$, $\phi_{NS}$, and $d_{NS}/d$. Since the $\xi_g$-scaling is

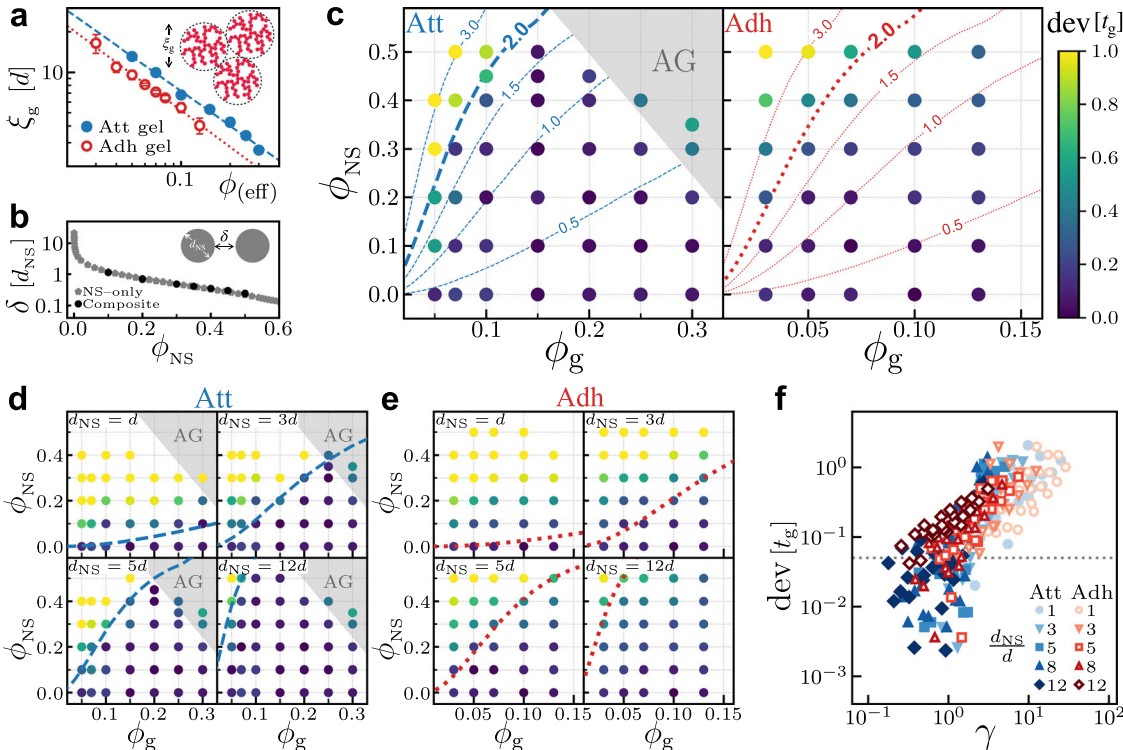

**Fig. 3 | Interplay between lengthscales affects gelation with NS particles.**
**a** Characteristic lengthscale $\xi_g$ as a function of volume fraction $\phi$ in colloidal gels. Dashed and dotted lines are power-law fittings with results shown in Eqs. (4) and (5). **b** Average spacing $\delta$ between NS particles as a function of $\phi_{NS}$. Gray stars are from individual simulations of only NS particles, while black filled circles are from binary composites with various $\phi_g$ and $d_{NS}$. The NS spacing $\delta$ is determined through Voronoi analysis in both cases (see the Methods section). **c** $\phi_g$–$\phi_{NS}$ diagrams in attractive (left) and adhesive (right) systems with $d_{NS} = 8d$. Color indicates the value of dev[$t_g$]. Gray region refers to AG regime with $\phi_{eff} > 0.4$. Lines refer to the iso-$\gamma$ lines with values shown on each of them. **d**, **e** are diagrams in attractive systems and adhesive systems with various $d_{NS}$ shown on the upper-left corner of each diagram. Dashed and dotted lines refer to $\gamma = 2$. **f** Plot of dev[$t_g$] versus $\gamma$, including all the data presented in **c**–**e**. Deviations below the dotted line (dev[$t_g$] = 0.05) are considered as random error only.

different in the attractive and adhesive gels, as shown in Eqs. (4) and (5), the lengthscale ratio $\gamma$ also depends on the specific contact model.

We find that the $t_g$ deviation from the master curve (Fig. 2c) correlates with $\gamma$. To quantify the degree of deviation, we use the distance between the average of measured $t_g$ and the predicted $t_g^{fit}$ from power-law fitting, defined as follows:

$$\text{dev}[t_g] \equiv |\log t_g - \log t_g^{fit}|, \tag{6}$$

where $t_g^{fit}$ refers to the gelation time calculated by Eq. (1) or (2) but using $\phi_{eff}$ instead of $\phi$. With all the data shown in Fig. 2c, we map out the $\phi_g$–$\phi_{NS}$ diagrams for both gel models at $d_{NS} = 8d$, Fig. 3c. The quantified deviation dev[$t_g$] is represented by the colormap.

For attractive systems, while most data show small dev[$t_g$], we observe two regions that present visible deviations in the diagram, Fig. 3c (left). At high $\phi_g$ and $\phi_{NS}$, the effective volume fraction $\phi_{eff}$ is so high that the system falls in the AG regime with $\phi_{eff} > 0.4$ (gray). This is consistent with the deviation at high $\phi_{eff}$ in Fig. 2c, where the measured $t_g$ falls below the power-law fitting (blue dashed line).

Significant deviation also occurs at low $\phi_g$ and high $\phi_{NS}$ (upper left corner of the diagram). By drawing the iso-$\gamma$ lines, we find that higher $\gamma$ leads to more prominent deviation dev[$t_g$]. Namely, when the characteristic size in gel $\xi_g$ far exceeds the spacing $\delta$ between NS particles, gelation is greatly hindered by their interplay, which dominates over the effect of $\phi_{eff}$.

We use the same method to calculate the ratio $\gamma$ as well as the deviation dev[$t_g$] in adhesive systems and find that this scenario appears to still work, Fig. 3c (right). As the lengthscale ratio $\gamma$ increases, the deviation becomes significant at low $\phi_g$ and high $\phi_{NS}$. For both systems, visually, the iso-$\gamma$ line of $\gamma = 2$ demarcates the regions with low

and high deviations. This result supports our argument that, as an important factor, the lengthscale interplay primarily affects the gelation process in binary systems at high $\gamma$.

The role of lengthscale ratio $\gamma$ is further verified by varying $d_{NS}$ from $d$ to $12d$ in both gel models, Fig. 3d, e. At the same composition, the lengthscale ratio $\gamma$ decreases with the NS particle size $d_{NS}$. For larger NS particles of $d_{NS} = 12d$, therefore, most data points fall on the master curve when plotting $t_g$ versus $\phi_{eff}$ (Supplementary Fig. 6), and $t_g$ deviation is greatly suppressed due to the small $\gamma$. As $d_{NS}$ decreases, deviation becomes increasingly significant. When the colloids and NS particles have comparable sizes (i.e., $d_{NS} = d$), almost all data points deviate from the master curve (Supplementary Fig. 6). Raw data of diagrams in Fig. 3d, e can be found in Supplementary Note 5.

Remarkably, the iso-$\gamma$ line at $\gamma = 2$ demarcates the low- and high-deviation regions in all cases, Fig. 3c–e. The positive correlation between deviation and lengthscale ratio $\gamma$ is obvious when plotting all the data together, Fig. 3f.

Regardless of the interaction (attractive or adhesive), composition ($\phi_g$ and $\phi_{NS}$), and particle size ratio $d_{NS}/d$, the lengthscale ratio $\gamma$, as well as the effective volume fraction $\phi_{eff}$, seems to characterize the gelation process in binary mixtures well.

## Growth of the largest cluster

The deviation from $\phi_{eff}$ scenario indicates an additional hindering effect at high $\gamma$. Such hindering results from the frustration of cluster growth. In particular, we examine the evolutions of particle fraction in the largest cluster $N_{lc}/N$. For each contact model, we compare systems with different compositions and $d_{NS}$ but the same $\phi_{eff}$ ($\phi_{eff} = 0.2$ for attraction and $\phi_{eff} = 0.1$ for adhesion), Fig. 4. The value of lengthscale ratio $\gamma$ is represented by the color. Compared with the pure gel at

$\phi = \phi_{eff}$ (black), binary systems with small $\gamma$ exhibit similar growth, while those with large $\gamma$ show a delayed increase in $N_{lc}/N$. Interestingly, the cluster morphology appears to be barely affected by NS particles regardless of $\gamma$ (Supplementary Note 6). These results explicitly show how the lengthscale interplay, represented by the unified parameter $\gamma$, affects colloidal gelation with NS particles.

### Behavior of NS particles

While colloids aggregate into a porous network as gelation proceeds, the dynamics of NS particles varies from system to system. At dilute $\phi_g$ and small $d_{NS}$, we expect NS particles to be able to diffuse even upon gelation since the pore size of the gel matrix is much larger than the NS particles. As the size of pores shrinks, the NS diffusion starts to be confined until finally 'locked' in the matrix cage as soon as a gel network is formed.

To probe the effect of $\phi_g$, $\phi_{NS}$, and $d_{NS}$ on NS dynamics, we measure the mean squared displacement (MSD) in various composite samples. For ease of comparison between colloids and NS, we normalize MSD by diffusion coefficient $D_{diff}$ for each species of the particles. We find that increasing $\phi_g$ and $d_{NS}$ both lead to the dynamical arrest of NS, which seems to occur simultaneously with that of colloids, Fig. 5a (left and middle). This may correspond to the case where NS size exceeds the pore size of the gel matrix, which decreases with $\phi_g$ as in Supplementary Fig. 4. We also notice that increasing $\phi_{NS}$ slows down the NS dynamics, Fig. 5a (right), probably due to the increasingly crowding surroundings[41].

Through radial distribution function (RDF), we also study the configuration of NS particles in binary composites. Without loss of generality, we use a specific composition of $\phi_g = 0.1$ and $\phi_{NS} = 0.3$ with four different $d_{NS}$. While a small peak appears at the second-nearest neighbor for small $d_{NS} = d$ and $3d$, such subtle spatial correlation does not present for larger $d_{NS}$, Fig. 5b. We do not identify any sign of

crystallization for all cases, and there is little variation in RDF before and after gelation. These results imply that the depletion effect[42,43] (and the consequent Casimir-like attraction[44]) between NS particles is neglectable.

## DISCUSSION

To better illustrate the effect of NS particles, here we consider two limits, Fig. 6. For infinitely-large NS particles ($d_{NS} \to \infty$), the colloids behave as a continuum which is geometrically confined between solid-wall boundaries, i.e., the surfaces of NS particles. Within the colloidal phase, the real volume fraction is $\phi_{eff}$ rather than $\phi_g$, and the diffusion, as well as aggregation, is purely mediated by the background solvent of viscosity $\eta_f$. As the gelation time is proportional to the Brownian time $\tau_B$, we expect the scaling to be $t_g \sim \eta_f (\phi_{eff})^\alpha$, where $\alpha$ refers to an interaction-dependent exponent.

At another limit with $d_{NS} \to 0$, the NS particles form a continuum background in which sticky colloids are distributed, Fig. 6 (right). In such a case, confinement for colloids is absent so that the gelation time scales with the absolute volume fraction $\phi_g$ rather than the effective one $\phi_{eff}$. The continuum background is essentially a hard-sphere suspension, whose viscosity $\eta_{NS}$ increases with the volume fraction of NS particles[45]. Though such suspension is not a simple Newtonian fluid of viscosity $\eta_{NS}$ in general, we may expect so because the situation is close to equilibrium[46]. In this sense, the gelation time then has the form $t_g \sim \eta_{NS}(\phi_g)^\alpha$. For the smallest $d_{NS} = d$ we investigate, the background viscosity $\eta_{NS}$ increases with $\phi_{NS}$ roughly in a Krieger–Dougherty manner[47] (Supplementary Note 7).

The above arguments can be generalized by using an effective background viscosity $\eta_{eff}$ and an effective colloidal volume fraction $\phi_{eff}^g$ (differing from $\phi_{eff}$ in Eq. (3)) as follows:

$$t_g \propto \eta_{eff} \times (\phi_{eff}^g)^\alpha. \tag{7}$$

The values of $\eta_{eff}$ and $\phi_{eff}^g$ at the two limits are shown in Fig. 6. The collapsed gelation time $t_g$ in systems of $d_{NS} = 12d$ validates $\phi_{eff}$ at large NS particles. For binary systems with $d_{NS} = d$ and the same $\phi_g = 0.1$, the identical structure factor $S(q)$ (Supplementary Fig. 3) and void distribution (Supplementary Fig. 4) suggest that the effective volume fraction $\phi_{eff}^g$ reduces to $\phi_g$ at small $d_{NS}$. Though this conflicts with the previous assumption ($\xi_g$–$\phi_{eff}$ scaling), using $\phi_g$ instead of $\phi_{eff}$ seems to make little difference in the iso-$\gamma$ line (Supplementary Note 3).

As $d_{NS}$ decreases from infinity to zero, therefore, we expect transitions in both $\eta_{eff}$ (from $\eta_f$ to $\eta_{NS}$) and $\phi_{eff}^g$ (from $\phi_{eff}$ to $\phi_g$). At intermediate $d_{NS}$, the interpenetration between NS particles and ramified clusters, which weakens the confinement effect and thereby decreases the effective volume fraction $\phi_{eff}^g$, becomes possible. Meanwhile, the further aggregation of colloidal clusters with size comparable to the NS spacing ($\gamma \sim 1$) requires the rearrangement of NS particles, which turns on the transition in background viscosity from $\eta_f$

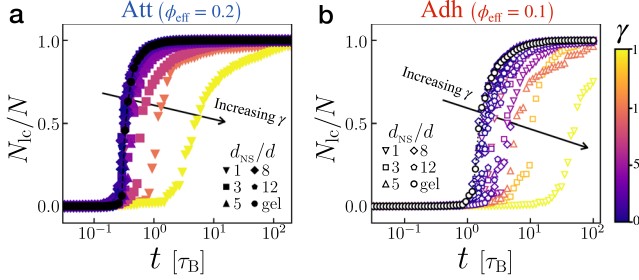

**Fig. 4 | Lengthscale ratio $\gamma$ controls cluster growth. a** Time evolutions of particle fraction in the largest cluster $N_{lc}/N$ of attractive systems with $\phi_{eff} = 0.2$. **b** Time evolutions of $N_{lc}/N$ in adhesive systems with $\phi_{eff} = 0.1$. In each plot, data of pure gels ($\phi = \phi_{eff}$) are shown in black. Detailed compositions are not shown; instead, we use $\gamma$ represented by the color. Arrows indicate increasing $\gamma$.

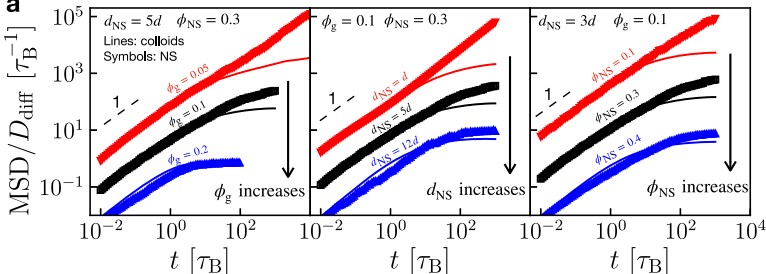
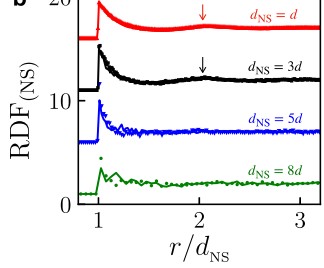
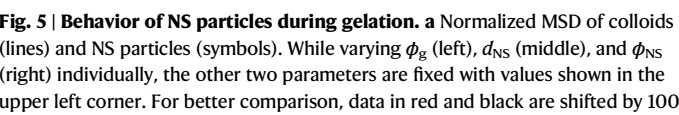

**Fig. 5 | Behavior of NS particles during gelation. a** Normalized MSD of colloids (lines) and NS particles (symbols). While varying $\phi_g$ (left), $d_{NS}$ (middle), and $\phi_{NS}$ (right) individually, the other two parameters are fixed with values shown in the upper left corner. For better comparison, data in red and black are shifted by 100

and 10, respectively. **b** RDF of NS particles in composites of $\phi_g = 0.1$ and $\phi_{NS} = 0.3$. Visible peaks are highlighted by arrows. Lines and symbols represent data before and after gelation. For better comparison, data of $d_{NS} = d$, $3d$, and $5d$ are shifted by 15, 10, and 5, respectively.

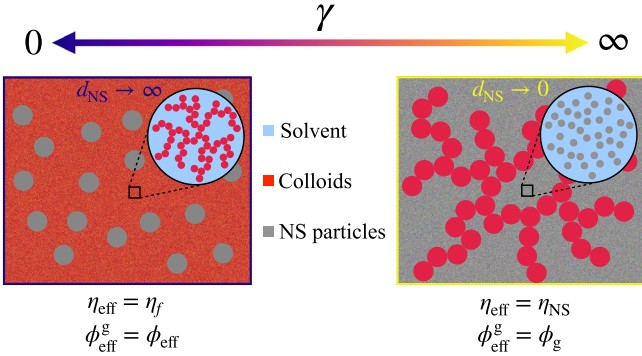

**Fig. 6 | Schematic illustration of two limit cases in binary systems.** Left: $d_{NS} \to \infty$ ($\gamma \to 0$). Right: $d_{NS} \to 0$ ($\gamma \to \infty$).

to $\eta_{NS}$. In particular, pinning NS particles during gelation leads to diverging gelation time $t_g$ beyond $\gamma = 2$ (Supplementary Note 8). In this way, the $t_g$ deviation occurs as a result of both the decrease in effective volume fraction $\phi_{eff}^g$ and the increase of background viscosity $\eta_{eff}$.

While the above discussion merely considers $d_{NS}$, the lengthscale ratio $\gamma$, including $\phi_g$, $\phi_{NS}$, and $d_{NS}$, offers a generic, unified measure in all binary composites. The lengthscale competition then essentially represents a mechanism transition between the two limit cases as shown in Fig. 6. While using global quantities in the expression of $\gamma$, we note that local details may also play a secondary role. For example, the lengthscale ratio $\gamma$ assumes uniform sizes for clusters and NS spacing, which, in practice, both have size distributions (Supplementary Note 4) and irregular morphologies (such as porosity and tortuosity[40,48]). Meanwhile, as binary systems involve different interactions, the coupling of localized glassy dynamics may also interfere with the formation of gel network[49]. These factors, as well as the others not listed, may more or less affect the gelation process and cannot be fully captured by a sole quantity $\gamma$. This is consistent with the scattering of data points in Fig. 3f.

In summary, we use Langevin dynamics simulations to investigate colloidal gelation with non-sticky particulate fillers. Through extensive exploration in the parameter space, we find that the interplay between two lengthscales ($\xi_g$ and $\delta$), represented by their ratio $\gamma$, matters in composite gelation. At $\gamma < 2$, the NS particles act as geometric confinement and effectively increase the colloidal concentration in the form of $\phi_{eff}$. As $\gamma$ increases, gelation is progressively hindered as a result of both the decrease in effective concentration and the increasingly-viscous background. Our results not only shed light on industrial formulation and processing, but also open up a new scheme for the tunability in practical gel materials. Though precise prediction is still challenging due to the missing of microscopic details, we successfully capture the generic importance of lengthscale competition in multi-component systems. Our finding will inspire the fundamental understanding and may lead to the efficient development of colloidal composite materials.

## METHODS

### Simulations

We perform Langevin dynamics simulations on LAMMPS[50]. Under thermostat at $k_B T$, our system contains two species of spherical particles which differ in size and interaction. The first species consists of sticky particles with an average diameter $d$ (bidispersed with $0.87d$ and $1.13d$ to prevent crystallization), while the second species consists of elastic spheres of diameter $d_{NS}$. To ensure that both colloids and NS particles are diffusive within the relevant time range for the gelation process ($\gtrsim 0.1\tau_B$), we set the particle mass proportional to the particle size with the constant damping time $m/3\pi\eta d \ll \tau_B$ (see Supplementary Note 9 for the details). The NS–NS and NS–g

interactions are simple elastic repulsions when overlapped, modeled by a modified Hertzian model with a high modulus $Ed^3 \gg k_B T$. To capture the interaction between sticky colloids, we use the Derjaguin−Muller−Toporov (DMT) contact model[51] with a sufficiently strong attraction $U_{att} = 20k_B T$. With the same modulus $E$, the overlap caused by cohesion is small ($\approx 0.01d$) at force balance, ensuring short-range attraction. Tangential constraints on sliding, rolling, and twisting (Fig. 1a) are all modeled in a modified Coulomb manner with the same spring constant and friction coefficient $\mu$. Respectively, we set $\mu = 0$ for attraction without constraints and $\mu = 1$ for adhesion with constraints on all three motions.

Our simulation occurs in a cubic box of side length $L = 50d$ with periodic boundaries, which is sufficiently large for bulk condition (Supplementary Fig. 10). In the absence of colloid−colloid attraction, an initial configuration is generated through multiple relaxations. We first randomize NS particles and wait for them to relax for $100\tau_B$, and then relax randomly-distributed, non-attractive colloids with pinned NS particles for another $100\tau_B$. Upon equilibrium, we unpin NS particles and allow the bulk system to relax shortly for $10\tau_B$. The initial state generated by such a pre-relaxing protocol exhibits no overlapping and no visible aggregation caused by depletion forces. We find little dependence on the pre-relaxing duration (Supplementary Fig. 11), suggesting the robustness of our results.

Starting from a homogeneous random configuration, each system evolves up to $10^4\tau_B$. We run each simulation for at least three times, with the data points and error bars shown in this work representing the average and standard deviation, respectively. Visualization and part of data analysis are carried out using OVITO[52].

### Determining gelation time

A robust criterion for gelation is crucial to accurately determine the gelation time $t_g$. The experimental convention views the liquid-to-solid transition, typically characterized by oscillatory rheology[53], as the gelation point. Inspired by the Maxwell criteria for stability[29], recent work, including both simulation[24] and experiment[31], correlates the evolution of clusters of isostatic particles (contact number $N \geq N_c$) with colloidal gelation and confirms the validity of such structural indicator by comparison with macroscopic rheology.

In this work, we define the gelation time $t_g$ as the time required for the isostaticity percolation. As Fig. 1b shows, we first extract all isostatic particles with $N \geq N_c$ (Table. 1 in ref. 28) and then examine their connectivity. Gelation is determined if there exist clusters percolating through periodic boundaries in all three directions ($x$, $y$, and $z$). Recent work correlates rigidity percolation with gelation boundary[54]. For adhesive systems, our method (isostatic percolation with $N_c^{adh} = 2$) gives the same result as rigidity percolation, since each pair constitutes a minimal rigid cluster. Yet this may not hold for attractive contacts with no tangential constraints, where 3D rigidity analysis is challenging[31] (and beyond the scope of this work). Therefore, we consistently apply the isostaticity method for attractive systems with $N_c^{att} = 6$.

### Voronoi analysis and NS-particle spacing

To measure the average spacing between NS particles $\delta$, we perform simulations of only NS particles at different volume fractions $\phi_{NS}$. Upon Brownian relaxation, Voronoi analysis is performed, and the spacing between each pair of particles $\delta_i$ is estimated as follows:

$$\delta_i = 2 \times \left( \frac{3V_{cell,i}}{4\pi} \right)^{1/3} - d_{NS}, \tag{8}$$

where $V_{cell,i}$ refers to the volume of the Voronoi cell of the $i$-th particle. Here we assume isotropic distribution and regard each Voronoi polyhedron as an equivalent sphere. Then the average spacing $\delta = \langle \delta_i \rangle$. The distribution of $\delta_i$ can be found in Supplementary Fig. 5.

## Data availability

The data generated in this study are provided in the Supplementary Information/Source Data file. Source data are provided with this paper.

## Code availability

The codes of the computer simulations are available from the corresponding authors upon reasonable request.

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

## Acknowledgements

R.S. acknowledge funding from Wenzhou Institute, University of Chinese Academy of Sciences (WIUCASQD2020002) and National Natural Science Foundation of China (12174390, 12150610463). Y.J. acknowledge funding from China Postdoctoral Science Foundation (2022M723114). We thank PostDoctoral Association (PDA) and Journal Club (JC) in Wenzhou Institute for fruitful discussions.

## Author contributions

Y.J. conceived the research and carried out the simulation. Y.J. and R.S. analyzed the data and wrote the manuscript.

## Competing interests

The authors declare no competing interests.
