## [Peer Review File · Nature Communications]

REVIEWER COMMENTS

Reviewer #2 (Remarks to the Author):

In this paper, Y. Jiang and R. Seto study numerically the effect of passive repulsive inclusions of different sizes on the gelation transition for two model systems consisting of colloidal sticky particles or a more constrained adhesion-like interaction. The main finding is that the repulsive particles induce structural correlations that lead to colloid confinement that would accelerate gelation. The size effect is rationalized in terms of competing length scales based on the effective porosity of the repulsive particles and the mesh size of the colloidal gel. The analysis is well done and the manuscript is well written. It can potentially be considered for publication, however, I am not convinced by the authors' interpretation (see my comments) and would like the authors to address my concerns before any further consideration.

1) My first comment is on the way the gelation boundary is detected. The Maxwell criterion is certainly a good first characterisation of the gelation, however, (i) the criterion is adapted in the case of 'adh' particles since more constraints are added to the interaction, can the authors elaborate more on the constraints counting, $N_c=2$ seems a bit strange to me, it means that in 3d a chain of colloids is marginally solid? or an open structure like a tree-like structure is solid? What about the 2d case? (ii) I am not convinced that the gelation would happen exactly at the isostatic point in these systems. This view is somehow a bit biased by the jamming transition that indeed coincides with a hyper to hypostatic transition. In gels, one probably needs to check the percolation of the rigid clusters. This has been shown recently to coincide with the gelation boundary see 'Correlated rigidity percolation and colloidal gels, Phys. Rev. Lett. 123, 058001 (2019).'

The authors mention that 'the above results justify our gel simulations as well as the gelation criterion we used.' based on figure 1. I think this statement can be misleading since the measure of the structure factor has nothing to do with the gelation criterion. I agree that they cross the gelation boundary but my previous point needs to be addressed to really conclude.

2) In the methods section, the authors do not mention anything about the mass of their particles. If the authors assume that their inclusions have the same mass density as the colloids, they need to adjust it. This is crucial to have the right long-time diffusion dynamics for their particles that would depend only on the particle diameter and not the mass.

3) The authors claims that Δ is time independent. This is probably true for short times. Did they let the SN particles to reasably diffuse before measuring? I am wondering if the authors results are robust if the system is let to evolve over time sufficiently large before switching the attractive interactions. Maybe memory effects of the initial configuration persist for large SN but not for small SN which would lead to deviation from their master curve.

4) Figure 4 is a bit misleading to me. I think the authors need to check system size effects while changing the volume fraction and comparing the size of the largest percolating cluster in order to conclude.

5) The ratio of the length scales is well correlated with deviations from the master curve. Can the authors propose a better scaling of their data into a corrected master curve that take into account this observation?

6) it would easier for the reader to highlight in figure 1d plots that are below gelation and above gelation.

Reviewer #3 (Remarks to the Author):

This paper presents simulation studies of gelation in binary mixtures of attractive/adhesive colloids (gel particles) and non-interacting hard sphere colloids (NS particles) as a function of various control parameters, including sizes and volume fractions of the two colloidal particle species.

One would expect that the primary effect of the non-interacting hard spheres would be to reduce the free volume of the system and thereby increase the effective volume fraction of the attractive/adhesive colloids. This is indeed the case in much of the parameter space of these systems (for both attractive and adhesive colloids), and the gel time (based on the percolation time of a network of of contacting particles satisfying an appropriate isostaticity condition) of the binary mixtures lies on a master curve defined by the gelation of a corresponding pure system of attractive/adhesive colloids using the effective volume fraction as a parameter.

Interestingly, at low volume fraction of gelling particles and relative high volume fractions of NS particles, the gel time is relatively slower than predicted by the master curve. This tendency is enhanced, at fixed volume fraction, when the size of the NS particles is not large compared to the size of the gel particles. The authors furthermore find a phenomenological explanation of this behaviour as resulting from a competition of two physical length scales: the average interparticle separation of the NS particles (as a function of their volume fraction) and the characteristic cluster size of the gelled particles (determined from the maximum in the static structure factor, $S(q)$, of the pure system at the gelation point), as a function of their effective volume fraction in the mixture.

Overall, this paper presents interesting (and possibly universal) results about the effects of neutral particles on gelation behaviour of an attractive colloidal suspension. The simulations seem to be carefully conducted and thorough. The paper is well written as well. I would support publication of this work in Nature Communications after the authors address some issues:

(1) It is a bit unfortunate that the gel length in the mixture could not be directly determined by the structure factor of the gel particles in the mixture. It seems to me that using the gel length for an equivalent pure system as a proxy for the the gel length in the mixture is an uncontrolled approximation. The authors need to do a better job in making this case. I also think that there is some useful physical information in the $S(q)$ data shown in Supplemental Figure 3a that could be conveyed, in particular about the sharpening of the peaks in the gel region with increasing volume fraction of NS particles.

(2) The paper focusses on the behaviour of the sticky particles. I would like to see more data about the behaviour of the NS particles. In addition to the data shown for NS particles in the supplemental document, it would be useful to show data on the mobility of the NS particles during the gelation process. Specifically, one could show the MSD vs time for the NS particles (as was done in Supplemental Figure 1 for the gel particles). It will be interesting to know to what extent their motion is also arrested at the gel point (due to their gelled surroundings). Likewise, it would be interesting to see the structure factor (or even better, the radial distribution function) for the NS particles, to see if there are any subtle spatial correlations among the NS particles in the gelled state.

(3) Intuitively, it is plausible that the morphology of the gel clusters at $\gamma > 1$ are perturbed at their peripheries by the NS particle neighbours. It would be useful, therefore, to look for such possible deviations in cluster morphology at NS interparticle length scales.

(4) These studies were conducted in the strongly attractive limit, $U_{att} \gg kT$. In such a case, I would expect that inter particle association would be essentially irreversible, and if so, that this gelation process would be non-ergodic in nature. Please comment on this. In many experimental conditions

U_{att} is only moderately larger in magnitude than the thermal energy scale. How would the authors expect the behaviour to change in such a case? Is the universality they find contingent upon being in the non-thermal limit of $U_{att} \gg kT$?

(5) The literature review is inadequate. There is a substantial number of simulation works on bimodal systems not mentioned here. Please rectify this situation in the revision.

Reviewer #4 (Remarks to the Author):

The paper by Jiang and Seto studies the gelation of sticky colloids in presence of non sticky fillers, a rather common situation in many industrial processes.

The authors first study the gelation behaviour of two models of sticky particles: the attractive model that show only centrosymmetric attraction and hard-sphere repulsion, and the adhesive model that add constraints on sliding, rolling and twisting. This last model is inspired from recent work on non Brownian suspensions.

They determine mechanical gelation time from a proxy criterion (percolation of particles with more contact than the isostaticity criterion) which relevance has been verified from mechanical measurements in the case of the attractive model and that they here extend to the adhesive model that has a different isostaticity criterion.

They also study the structure of the gel network, distinguishing the DLCA fractal for the adhesive model, and the two-lengthscale structure typical of arrested phase separation for the attractive model.

They then proceed to simulate mixtures of one or the other of the two previous gelling systems on one hand and non sticky (NS) particles on the other hand. Following the literature, they first model the effect of such inert fillers as a reduction in the volume available to the gel, thus an effective volume fraction of the gel. They show that this model describes well the gelation time for large NS particles at high volume fraction of sticky particles, but fails at low volume fraction of sticky particles.

Furthermore, they identify two competing length scales that control this deviation: the correlation length of the gel at gelation time, and the average distance between NS particles. By varying the size of the NS particles, they convincingly show that the ratio between these two length scales is well correlated with the deviation to the effective volume fraction model.

In addition, they evidence that the growth of the largest cluster is more delayed by smaller NS particles, with respect to large NS particles at the same volume fraction.

Finally, they discuss limit cases theoretically. In particular, in the small NS particle limit, they assign the effect of NS particles to an effective volume fraction. The NS particles need to move to let the network form. Interestingly, supplementary data show that pinning the NS particles, effectively making the effective viscosity infinite for the gelling system, can completely prevent gelation.

Overall, I find that the manuscript is clearly written, lays a strong argumentation and the proper level of details to give all the necessary ingredients for reproducibility. I particularly like that both attractive and adhesive models follow in parallel until the end of the paper, backing up the claim of universality of the story. The final figure and discussion is also a strong point that speaks in a pedestrian way to non specialists.

Although the hard sphere model is an oversimplification of fillers seen in real processes, the present manuscript sets an important milestone in the study of gel+filler systems and will surely be used a reference for further studies.

I have a few remarks that may lead to possible improvements of the manuscript.

First, it is well known that the presence of excluded volume around objects can cause effective depletion interactions. For instance Royall, Louis and Tanaka (J. Chem. Phys 2007 <https://doi.org/10.1063/1.2755962>) show the effective depletion interaction between large hard spheres in a sea of smaller hard spheres, as predicted by Roth and coworkers (<https://doi.org/10.1103/PhysRevE.62.5360>, <https://doi.org/10.1063/1.1908765>). In practice, the pair distribution function between the larger HS oscillate with a period that is the size of the smaller HS: It is more favorable to fit an integer number of HS. Since the decay of these oscillations is quick, they should have a measurable effect only when the large objects are close together, corresponding typically to $\Delta/d < 5$ in the present problem. If I am reading Fig3.b correctly, it means $\phi_{NS} > 0.3$ in the case $d_{NS} = 8d$. Incidentally, this might correspond to the regime of high deviation of the gelation time. It might be worth adding a few more "Composite" points on Fig 3b to check whether Δ becomes quantified at high ϕ_{NS} , and whether this has an effect on $\text{dev}[t_g]$.

Furthermore, when the smaller objects are aggregating, depletion or Casimir-like forces are observed with ranges scaling like the size of the largest cluster, see Gnan, Zaccarelli and Sciortino (Nature Com. 2014 <https://doi.org/10.1038/ncomms4267>). Therefore, one would expect that the

configuration of NS spheres is affected on even larger length scale, e.g. at lower ϕ_{NS} . I think that Fig3b answers well that issue, but it might be worth checking Gnan et al predictions.

In the same line of thoughts, I am surprised that it is possible to neglect the exclusion shell around NS particles in Eq2. Is the collapse in Fig2c any better if the exclusion shell is taken into account?

Same question for Fig3f, where the correlation between $\text{dev}[t_g]$ and γ seems to be lost around $\gamma=1$. Is it fixed by taking the exclusion shell into account, e.g. $\gamma = \xi_g/(\delta-d)$ instead of ξ_g/δ ?

Have you tested expressions in the literature for $\eta_{NS}=\eta(\phi_{NS})$ to confirm that the gelation time in the small d_{NS} regime scales as $\eta_{NS} * \phi_g^\alpha$?

Finally, I spotted a typo and suggest two formal corrections:

- p51322 "the counter line" => "the contour line"
- Maybe rather than "contour line", "the iso-gamma line" would be more understandable?
- In the caption of Fig4, I suggest to change "varies over time" to "time evolution" (twice).

NCOMMS-22-48521: Response to Reviewers

We thank the reviewers for their thoughtful comments and are pleased that they find our results interesting and consider publication with revisions. We have given careful consideration to their concerns and carried out additional simulations and analyses to strengthen our study. Please find below our point-to-point response to each reviewer, with their comments labeled in blue (#2), red (#3) and teal (#4), respectively. Corresponding revisions in the manuscript and Supplementary Information follow the same color scheme. We believe that the revised manuscript and our response should address all their questions and concerns.

Reviewer #2

1) My first comment is on the way the gelation boundary is detected. The maxwell criterion is certainly a good first characterization of the gelation, however, (i) the criterion is adapted in the case of ‘adh’ particles since more constraints are added to the interaction, can the authors elaborate more on the constraints counting, $N_c = 2$ seems a bit strange to me, it means that in $3d$ a chain of colloids is marginally solid? or an open structure like a tree like structure is solid? What about the $2d$ case? (ii) I am not convinced that the gelation would happen exactly at the isostatic point in these systems. This view is somehow a bit biased by the jamming transition that indeed coincides with a hyper to hypostatic transition. In gels, one probably need to check the percolation of the rigid clusters. This has been shown recently to coincide with the gelation boundary, see ‘*Correlated rigidity percolation and colloidal gels*’ [1]. The authors mention that ‘*the above results justify our gel simulations as well as the gelation criterion we used.*’ based on figure1. I think this statement can be misleading since the measure of the structure factor has nothing to do with the gelation criterion. I agree that they cross the gelation boundary but my previous point need to be addressed to really conclude.

We thank the reviewer for the thoughtful comment. We adopt such method from literature which have tested its validity through rheology in both experiments [2] and simulations [3]. The literature raised by the reviewer [1] offers an interesting insight and applies a pebble game algorithm for 2D rigidity analysis. Yet as already pointed out in [2], direct generalization to 3D is challenging [4]:

... Whether a network is rigid can be determined using a pebble game algorithm, but this method is limited to two-dimensional systems ...

Below we will discuss the cases in two contact models separately.

For adhesive contact where all relative motions are constrained, each pair of particles in contact constitutes a minimal rigid cluster so that geometric percolation (i.e., isostatic percolation with $N_c = 2$) is equivalent to rigidity percolation. In principal, as long as it percolates, we do expect a chain or a tree-like structure of adhesive particles to be capable of being a marginal solid. In 2D, the twisting mode is absent and constraint counting still gives $N_c = 2$. The derivation of N_c in different cases can be found in [5] (note different symbols).

The case of attractive contact without tangential constraints is more complicated, and we do agree that the isostatic percolation may not always refer to the gelation point exactly. Here we use an alternative method for rigidity analysis, which has been verified to offer similar results with the pebble game in 2D [6]. Briefly, this method starts from the dynamical matrix and identifies rigid bonds based on the zero-mode displacements (see Supplementary material of [6] for more details). By analyzing pure gels at three different concentrations, we

Response Fig. 1. Comparison of two methods of gelation determination. **a** Snapshots of $\phi = 0.05$ gel at gelation points determined by two methods. Different clusters are labeled by different colors, while light pink represents either particles with $N < N_c = 6$ (isostaticity, left) or isolated particles without rigid bonds connected to themselves (rigidity, right). **b** Gelation time t_g from the two methods at three different volume fractions ϕ . Dashed lines refer to power-law fittings.

do not find significant difference in the gelation point determination, Response Fig. 1. Compared with the original method (isostaticity), this new method (rigidity) is time-consuming (≈ 1 day for 1 sample) and may detract from our main scope. Therefore, we still apply the isostaticity method in this work, with possible concerns discussed in the Methods section (Page 8, line 542–550).

As the reviewer stated, $S(q)$ indeed does not correlate with the gelation criterion we used. The original text “*the above results ... gelation criterion we used*” actually refers to the power-law exponents of measured t_g , which roughly agree with the values in the literature. We accordingly revised the manuscript for better clarification (Page 2, line 134–137; Page 3, line 170–172).

2) In the methods section, the authors does not mention anything about the mass of their particles. If the authors assume that their inclusion have same mass density as the colloids, they need to adjust it. This is crucial to have the right long time diffusion dynamics for their particles that would depend only on the particle diameter and not the mass.

We use the Langevin equation to implement the Brownian motion of colloids. According to the fluctuation–dissipation theorem, the long-term diffusion coefficient $D_{\text{diff}} = \frac{k_B T}{3\pi\eta d}$ depends only on the particle diameter d . By contrast, the mass m introduces a damping timescale $t_{\text{damp}} = \frac{m}{3\pi\eta d}$ below which particles behave in a ballistic manner. To illustrate, we simulate $N = 5000$ non-interacting colloids and measure their ensemble-averaged mean square displacements (MSD) at different m . Increasing the mass m broadens the inertial

regime ($\text{MSD} \propto t^2$) while the long-term diffusion is barely affected, Response Fig. 2 a.

Response Fig. 2. Mass effect on colloidal dynamics. **a** Time evolution of MSD at different masses. Dashed lines refer to damping times t_{damp} , below which ballistic behaviors ($\text{MSD} \propto t^2$) manifest. **b** Time evolution of MSD at different sizes d , with $m_i \propto d_i$ used in this work. Solid line represents consistent t_{damp} , which is well separated from our interested time range (gray). **c** Different mass scalings (open: $m_i \propto d_i$; solid: $m_i \propto d_i^3$) have little effect on the gelation time t_g . Here we use composites with large NS $d_{\text{NS}} = 8d$.

For pure gel simulations, we consider four timescales in ascending order: time step dt , damping time t_{damp} , interaction time unit $t_{\text{inter}} = \sqrt{\frac{md^2}{U_{\text{att}}}}$ [7] and Brownian time τ_B . We expect $dt \ll t_{\text{damp}} \ll t_{\text{inter}} \ll \tau_B$ with reasons listed below:

$$dt \ll t_{\text{damp}} : \text{stable simulation}$$

$$t_{\text{damp}} \ll t_{\text{inter}} : \text{overdamped condition}$$

$$t_{\text{damp}} \ll \tau_B : \text{diffusivity within gelation timescale}$$

$$t_{\text{inter}} \ll \tau_B : \text{stiff particles.}$$

In particular, we set $dt = 2 \times 10^{-2} t_{\text{damp}} = 1.2 \times 10^{-3} t_{\text{inter}} = 2 \times 10^{-5} \tau_B$, where timescales are well separated and the computing cost ($\propto \tau_B/dt$) remains appropriate.

The case in binary composite is more complicated, as we simulate with large size ratio of NS to colloids ($d_{\text{NS}}/d = 1$ to 12) and the size difference is expected to be reflected in different mass. If simply applying constant mass density (i.e., $m_i \propto d_i^3$), the range of t_{damp} and t_{inter} will be broadened so that the separation of timescales requires higher τ_B (i.e., higher computing cost). Under this context, we apply $m_i \propto d_i$ to set a constant t_{damp} for all particles and $t_{\text{damp}} = 10^{-3} \tau_B \ll \tau_B$ to ensure that both colloids and NS particles are diffusive within the relevant time range for gelation process ($\geq 0.1\tau_B$), Response Fig. 2 b.

In fact, for the parameters used in this work, $m_i \propto d_i$ and $m_i \propto d_i^3$ make small difference in the results, Response Fig. 2 c. This is probably because the slow motion (either ballistic or diffusive) of NS particles barely affects gelation. Yet to ensure correct physical process,

we apply $m_i \propto d_i$ throughout this work. We added relevant discussions to the Methods section (Page 7, line 485–489) as well as Supplementary Note 9.

3) The authors claims that δ is time independent. This is probably true for short times. Did they let the NS particles to reasonably diffuse before measuring? I am wondering if the authors results are robust if the system is let to evolve over time sufficiently large before switching the attractive interactions. Maybe memory effects of the initial configuration persist for large NS but not for small NS which would lead to deviation from their master curve.

Response Fig. 3. Independence of pre-relaxing time t_{relax} . We use an attractive system of $\phi_g = 0.1$, $\phi_{\text{NS}} = 0.3$ and $d_{\text{NS}} = 5d$, and vary the pre-relaxing time t_{relax} before switching on the colloidal attraction.

We apply pre-relaxing protocol before switching on the colloidal attraction. To confirm the independence on the pre-relaxing duration t_{relax} , we use an attractive system with $\phi_g = 0.1$, $\phi_{\text{NS}} = 0.3$ and $d_{\text{NS}} = 5d$, which has $\gamma > 2$ and exhibits visible deviation. Besides $t_{\text{relax}} = 10\tau_B$ used in this work, we vary t_{relax} over a wide range ($1\tau_B$ to $5000\tau_B$), but do not observe systematic dependence on t_{relax} for both NS spacing δ and gelation time t_g , Response Fig. 3. We added relevant discussion to the Methods section (Page 8, line 516–518) and Supplementary Note 9.

4) Figure 4 is a bit misleading to me. I think the authors need to check system size effects while changing the volume fraction and comparing the size of the largest percolating cluster in order to conclude.

The largest cluster in Figure 4 is already normalized in the form of N_{lc}/N , where N refers to the total number of colloids. To better answer the reviewer’s concern, we carry out simulations of three individual systems (with the same $\phi_{\text{eff}} = 0.2$) at three box sizes ($L_{\text{box}}/d = 35, 50, \text{ and } 70$). The information of each system is shown in Response Fig. 4, in which we hardly see the system-size dependence around the system size we investigated

Response Fig. 4. System size independence. We measure the largest cluster (lc) growth in three different systems (a gel and two binary mixtures, see detailed compositions on the top of each graph) with three different box sizes. Lines represent gelation point t_g .

($L_{\text{box}}/d = 50$). To make our results more robust, we added this result to the Methods section (Page 8, line 505–506) and Supplementary Note 9.

5) The ratio of the length scales is well correlated with deviations from the master curve. Can the authors propose a better scaling of their data into a corrected master curve that take into account this observation?

Though well correlation is present between $\text{dev}[t_g]$ and γ , the data in Figure 3f are quite scattered visually. As shown in Figure 6, the deviation with larger $\gamma \rightarrow \infty$ results from 1) decrease in the real effective concentration ϕ_{eff}^g and 2) increase in the background viscosity η_{NS} . These two transitions may occur asynchronously. Therefore, while γ solely offers a robust but rough prediction, we may expect multiple quantities to well characterize the deviation $\text{dev}[t_g]$. Moreover, the lengthscale ratio γ only represents a global tendency. It does not take local details (such as distributions of ξ_g and δ), which may play a role in gelation, into account. We, therefore, do not expect a better scaling through simple correction based on γ .

6) It would be easier for the reader to highlight in figure 1d plots that are below gelation and above gelation.

Thanks for the suggestion. We highlighted the data before and upon gelation using open and solid symbols respectively, see new Figure 1d.

Reviewer #3

1) It is a bit unfortunate that the gel length in the mixture could not be directly determined by the structure factor of the gel particles in the mixture. It seems to me that using the gel length for an equivalent pure system as a proxy for the gel length in the mixture is an uncontrolled approximation. The authors need to do a better job in making this case. I also think that there is some useful physical information in the $S(q)$ data shown in Supplementary Figure 3a that could be conveyed, in particular about the sharpening of the peaks in the gel region with increasing volume fraction of NS particles.

We thank the reviewer for the thoughtful comments and suggestions. Structural factor $S(q)$, corresponding to experimental small-angle scattering, is a canonical characterization for colloidal gel structure [8]. Yet just as the reviewer mentioned, it unfortunately cannot be used to extract the gel length in binary composites, even though we only consider the colloid–colloid structure factor $S_{cc}(q)$. As Supplementary Fig. 3a shows, pores created by large NS particles inevitably manifest at a large lengthscale (low q) and coincide with the gel length. In such cases, disentanglement of lengthscales is challenging. We expect the sharpening of peaks in the gel region to directly result from NS particles, as the peak position roughly agrees with the NS size d_{NS} . Similar sharpening of $S(q)$ peaks is also observed in hard-sphere systems with increasing concentration, such as Figure 1 in [9].

Response Fig. 5. Void length distribution. **a** Schematic method to determine void distribution. **b** Probability density function (PDF) of void length ξ_{void} in attractive gels. **c** Distribution of ξ_{void} in colloidal gels (black) and binary composites with $\phi_g = 0.1$ and $\phi_{NS} = 0.3$ (so that $\phi_{\text{eff}} = 0.143$).

To justify our assumption, here we use an alternative method, adopted from [10], to account for the lengthscale within the gel matrix. We first divide the space into cubic grids with the mesh much smaller than particles ($d_{\text{mesh}} = 0.1d$). For each lattice point, we define the void volume V_{void} as the volume of a sphere in contact with the nearest-neighbor particle, with the center of sphere at the lattice point, Response Fig. 5a. A void lengthscale $\xi_{\text{void}} = \sqrt[3]{V_{\text{void}}}$ can then be counted. For attractive colloidal gels, the distribution of ξ_{void} varies with the volume fraction ϕ , and the peak position (corresponding to a characteristic

void length) shifts to lower as ϕ increases, Response Fig. **5 b**. This is consistent with ξ_g in Figure **3a**.

For binary composites, we exclude the lattice points inside NS particles so that only the voids within gel matrix are counted. Here we compare composites with $\phi_g = 0.1$ and $\phi_{NS} = 0.3$ (so that $\phi_{\text{eff}} = 0.143$) and gels with $\phi = 0.1$ and $\phi = 0.143$, respectively. At small $d_{NS} = d$, the ξ_{void} distribution of binary composite is almost identical to that of $\phi = \phi_g = 0.1$ gel, consistent with the collapsed structure factors in Supplementary Fig. **3b**. At large $d_{NS} = 8d$, by contrast, the peak position of composite coincides with a gel at $\phi = \phi_{\text{eff}} = 0.143$, Response Fig. **5 c**. This suggests that the gel length in a mixture is comparable to that in a pure gel system with $\phi = \phi_{\text{eff}}$. The small difference in the distribution may stem from the spherical boundary of NS particles.

These results justify our assumption, i.e., *using the gel length for an equivalent pure system as a proxy for the gel length in the mixture* is reasonable. Though there is a variation from ϕ_{eff} to ϕ_g for different NS sizes, we have shown that this has little effect on the iso- γ curve (Supplementary Fig. **3c**). We added relevant discussions to the manuscript (Page 5, line 247–250; Page 7, line 419–420) and Supplementary Note 3.

2) The paper focuses on the behavior of the sticky particles. I would like to see more data about the behavior of the NS particles. In addition to the data shown for NS particles in the Supplementary document, it would be useful to show data on the mobility of the NS particles during the gelation process. Specifically, one could show the MSD vs time for the NS particles (as was done in Supplementary Figure 1 for the gel particles). It will be interesting to know to what extent their motion is also arrested at the gel point (due to their gelled surroundings). Likewise, it would be interesting to see the structure factor (or even better, the radial distribution function) for the NS particles, to see if there are any subtle spatial correlations among the NS particles in the gelled state.

If ignoring the variation of gel structure during gelation, then we may expect NS particles to diffuse within an porous matrix. In this sense, the ratio of pore size d_{pore} to NS size d_{NS} is important. At $d_{\text{pore}} \ll d_{NS}$, NS particles are dynamically arrested once gelation completes since further diffusion requires the rearrangement of the gel network, which is energy costly. On the other hand, where $d_{\text{pore}} \gg d_{NS}$, NS particles actually diffuse within an infinitely-dilute network so that diffusive behavior persists even upon gelation.

To confirm the above argument, we further investigate the NS behavior at different systems during gelation, Response Fig. **6 a–c**. For ease of comparison, we measure the MSDs of colloids and NS particles separately and normalize them by the diffusion coefficient D_{diff} of each species. While the dynamics of attractive colloids are gradually arrested as gelation proceeds, we notice that the long-term diffusions of NS particles are suppressed as ϕ_g and ϕ_{NS} increases, Response Fig. **6 a, b**. These results are consistent with our argument above,

since the characteristic pore size inside a gel matrix shrinks as concentration increases (Response Fig. 5b), so that the ratio $d_{\text{pore}}/d_{\text{NS}}$ decreases with ϕ_g and d_{NS} . The NS diffusion also varies with ϕ_{NS} , Response Fig. 6c, where long-term diffusion becomes slow as ϕ_{NS} increases. Apart from the increase in effective gel concentration, this result also agrees with the slowing-down dynamics in concentrated hard-sphere suspensions [11].

Response Fig. 6. Behavior of NS particles during gelation. a–c Mean squared displacements (MSD) of colloids (lines) and NS particles (symbols) during gelation with varying ϕ_g , d_{NS} and ϕ_{NS} . For ease of comparison, we normalize MSD by diffusion coefficient D_{diff} and shift the black and red curves by factors of 10 and 100, respectively. d NS–NS radial distribution function $g_{\text{NS}}(r)$ in composites of $\phi_g = 0.1$ and $\phi_{\text{NS}} = 0.3$. Visible peaks are highlighted by arrows. Lines and symbols represent data before and after gelation. For better comparison, data of $d_{\text{NS}} = d$, $3d$, $5d$ are shifted by 15, 10, and 5, respectively.

Besides dynamics, we also measure the radial distribution function (RDF) of NS particles $g_{\text{NS}}(r)$. Without loss of generality, we use a specific composition of $\phi_g = 0.1$ and $\phi_{\text{NS}} = 0.3$ with four different d_{NS} . While a small peak appears at the second-nearest neighbor for small $d_{\text{NS}} = d$ and $3d$, such subtle spatial correlation does not present for larger d_{NS} , Response Fig. 6d. For all cases, we do not identify any sign of crystalline structure, and there is only

little variation in RDF before and after gelation. We thereby expect the NS configuration to play a minor role in composite gelation.

We added these results and relevant discussions to the manuscript as a new subsection (Page 6, line 350–380 and Figure 5).

3) Intuitively, it is plausible that the morphology of the gel clusters at $\gamma > 1$ are perturbed at their peripheries by the NS particle neighbors. It would be useful, therefore, to look for such possible deviations in cluster morphology at NS interparticle length scales.

We do expect the existence of NS particles to affect the gel structure. However, we note that NS particles are always present during gelation so that their impact may not be just located on the periphery, which corresponds more to the case with inserted NS particles upon matrix formation. Yet admittedly, NS particles play a role in the colloidal diffusion and thereby may lead to clusters with higher anisotropy.

Response Fig. 7. Asphericity of gel clusters before gelation. For different contact models (Att – left; Adh – right), we compare a gel at $\phi = 0.07$ and two composites of $\phi_g = 0.07$ and $\phi_{NS} = 0.3$, with different NS sizes corresponding to $\gamma \approx 2$ ($d_{NS} = 8d$) and $\gamma \approx 5.3$ ($d_{NS} = 3d$) respectively. For each sample, we pick the frame at around $t \approx t_g/2$ and analysis over 1000 clusters composed of more than 10 colloids.

To investigate the cluster morphology, we use a dimensionless quantity asphericity (adapted from [12–14]) defined as:

$$\text{Normalized Asphericity (NA)} = \frac{\lambda_k^2 - \frac{1}{2}(\lambda_i^2 + \lambda_j^2)}{R_g^2}, \quad (1)$$

where $R_g = \sqrt{\lambda_i^2 + \lambda_j^2 + \lambda_k^2}$ refers to the gyration radius and $\lambda_{i,j,k}$ to the eigenvalues (ascending order) of the gyration tensor. In particular, NA= 0 represents spherically symmetric morphology while NA= 1 represents a linear chain of particles. We compare its distribution in gels ($\phi = 0.07$) and binary composites ($\phi_g = 0.07$ and $\phi_{NS} = 0.3$) at different NS sizes,

Response Fig. 7. For each sample, we pick the snapshot at $t \approx t_g/2$ and count for over 1000 clusters consisting more than 10 particles. For composites either at the critical boundary ($d_{\text{NS}} = 8d \rightarrow \gamma \approx 2$) or beyond it ($d_{\text{NS}} = 3d \rightarrow \gamma \approx 5.3$), we do not observe significant difference in the cluster asphericity.

According to the results above, we believe that the NS particles affect gelation without great distortion on the cluster morphology. We added relevant discussions to the manuscript (Page 6, line 344–346) and Supplementary Note 6.

4) These studies were conducted in the strongly attractive limit, $U_{\text{att}} \gg k_B T$. In such a case, I would expect that inter particle association would be essentially irreversible, and if so, that this gelation process would be non-ergodic in nature. Please comment on this. In many experimental conditions U_{att} is only moderately larger in magnitude than the thermal energy scale. How would the authors expect the behavior to change in such a case? Is the universality they find contingent upon being in the non-thermal limit of $U_{\text{att}} \gg k_B T$?

Response Fig. 8. Gelation with weak attraction ($U_{\text{att}} = 5.4k_B T$). **a** Gelation time t_g as a function of volume fraction ϕ . **b** Lengthscale ξ_g as a function of ϕ . Inset is a snapshot of a gel at $\phi = 0.1$. **c, d** Gelation time t_g versus ϕ_{eff} at two different NS sizes. **e, f** Deviation diagram of the data in **c** and **d**. We apply the same colorbar (not shown) to scale $\text{dev}[t_g]$. Dashed lines are iso- γ curves calculated from ξ_g in **b** and δ in Figure 3b.

The gelation process we simulate is indeed non-ergodic, typically corresponding to nanoparticles with strong attractions (e.g., van der Waals and hydrophobicity) in practice. As mentioned by the reviewer, the attraction magnitude in some experimental model

systems (e.g., depletion gels) is more moderate. In such cases, the colloidal phase separates into colloid-poor and colloid-rich domains until the latter is dynamically arrested (glass transition) into a bicontinuous, percolated texture. Such a gelation route (i.e., arrested spinodal decomposition [8]) is also non-ergodic.

Despite different scaling laws in lengthscale and gelation time, we expect similar scenario (competing lengthscales) to still apply at intermediate attraction. To verify, we further investigate attractive gels and counterpart composites with $d_{\text{NS}} = 8d$ and $3d$ at $U_{\text{att}} = 5.4k_{\text{B}}T$. Using the same analysis, we again observe deviation from ϕ_{eff} -prediction at high γ , Response Fig. 8, with the critical γ_c varying from 2 to 5.

By implementing an interaction-dependent γ_c , the results above appear to support the universality of our finding. Yet we note that as particle bonding becomes reversible, the gelation criterion we used may be improper as cluster breakage is possible upon isostaticity percolation.

5) The literature review is inadequate. There is a substantial number of simulation works on bimodal systems not mentioned here. Please rectify this situation in the revision.

We thank the reviewer to point this out. We added [15–18] as bimodal reference in the manuscript (Page 9, line 605–618). Please let us know if there is any benchmark work we omit.

Reviewer #4

1) First, it is well known that the presence of excluded volume around objects can cause effective depletion interactions. For instance Royall, Louis and Tanaka [19] show the effective depletion interaction between large hard spheres in a sea of smaller hard spheres, as predicted by Roth and coworkers [20, 21]. In practice, the pair distribution function between the larger HS oscillates with a period that is the size of the smaller HS: It is more favorable to fit an integer number of HS. Since the decay of these oscillations is quick, they should have a measurable effect only when the large objects are close together, corresponding typically to $\delta/d < 5$ in the present problem. If I am reading Fig3.b correctly, it means $\phi_{\text{NS}} > 0.3$ in the case $d_{\text{NS}} = 8d$. Incidentally, this might correspond to the regime of high deviation of the gelation time. It might be worth adding a few more “Composite” points on Fig 3b to check whether δ becomes quantified at high ϕ_{NS} , and whether this has an effect on $\text{dev}[t_g]$.

We thank the reviewer for the thoughtful comments. We measure the pair distribution function of NS $g_{\text{NS}}(r)$ in various systems and do not observe oscillation with d period, as shown in Response Fig. **6 d** ($\delta/d < 5$). Actually, if such effect plays a role in gelation, we may expect the boundary between low- and high-deviation regions to be insensitive to ϕ_g as colloid size d does not change. But this is not true.

Meanwhile, the little variation in $g_{\text{NS}}(r)$ during gelation indicates that the effective depletion between NS seems to be neglectable. Differing from the case in [20, 21], the smaller colloids in our system aggregate into larger clusters, which no longer contribute to the depletion once exceeding d_{NS} . That is, the depletant size increases over time. Hence depletion effect only manifests at the beginning of gelation, where colloids are mostly isolated.

We added more “Composite” points in Figure **3b** (in which we do not observed quantized δ) and relevant discussions about exclusion shell (Page 3, line 190–192).

2) Furthermore, when the smaller objects are aggregating, depletion or Casimir-like forces are observed with ranges scaling like the size of the largest cluster, see Gnan, Zaccarelli and Sciortino [22]. Therefore, one would expect that the configuration of NS spheres is affected on even larger length scale, e.g. at lower ϕ_{NS} . I think that Fig3b answers well that issue, but it might be worth checking Gnan et al predictions.

The literature [22] raised by the reviewer predicts an effective long-range Casimir-like attraction between NS particles when colloids aggregate. For a close-to-percolation system with $\phi_g = 0.1$ and $d_{\text{NS}} = 8d$, simple estimation (based on Eq. (6) in [22]) gives $U_{\text{Casimir}} \approx 4k_{\text{B}}T$ at contact. This magnitude may vary as ϕ_g or d_{NS}/d changes, yet the little variation in $g_{\text{NS}}(r)$ (Figure **5b**), as well as the time independence in δ (Supplementary Fig. **5a**, inset) and the consistency between NS-only data and composite data in Figure **3b**, suggests a minor role of such effect in our case. We added relevant discussions to the manuscript (Page 6, line 381–

383).

3) In the same line of thoughts, I am surprised that it is possible to neglect the exclusion shell around NS particles in Eq2. Is the collapse in Fig2c any better if the exclusion shell is taken into account?

Unlike the case in [20, 21], the smaller colloids in our composites are aggregating into larger clusters. We therefore expect exclusion shell to only apply at the initial stage of gelation, Response Fig. 9a (top). As clusters become larger and more ramified, it is possible for two clusters to further aggregate while accommodating (rather than excluding) NS particles, Response Fig. 9a (bottom). In such interpenetrating case [23], the concept of exclusion shell is not applicable.

Response Fig. 9. Exclusion shell effect. a Sketch to illustrate when exclusion shell applies and does not apply. b Replotted Figure 2c with exclusion shell considered.

So far, there is no accurate formula to account for the free volume subtracting exclusion shell. Here we use an approximated form (Eq. (9) in [24]) to generate a new ϕ_{eff} and replot Figure 2c with t_g versus the new ϕ_{eff} , Response Fig. 9b. Visually, such treatment makes the data more scattered (cf. Figure 2c).

4) Same question for Fig3f, where the correlation between $\text{dev}[t_g]$ and γ seems to be lost around $\gamma = 1$. Is it fixed by taking the exclusion shell into account, e.g., $\gamma = \xi_g / (\delta - d)$ instead of $\gamma = \xi_g / \delta$

As stated in the last response, we do not expect the exclusion shell effect to manifest as clusters grow. For the plot of $\text{dev}[t_g]$ versus γ , taking the exclusion shell into account does not produce better correlation, Response Fig. 10a. Actually, the scattering data around $\gamma = 0.05$ represents random error, which looks significant on the logarithmic scale. For pure gels, the measured t_g appears to well agree with the power-law fitting (Figure 1c), yet the deviation $\text{dev}[t_g]$ is non-zero and fluctuates around 0.05, Response Fig. 10b. We hence

simply regard $\text{dev}[t_g] < 0.05$ as zero, i.e., little deviation from the fitting result. For better illustration, we added a baseline $\text{dev}[t_g] = 0.05$ to Figure 3f.

Response Fig. 10. Scattering data in Figure 3f. **a** Replotted Figure 3f with exclusion shell considered in $\gamma = \xi_g/(\delta - d)$. **b** Deviation from power-law fitting $\text{dev}[t_g]$ in pure gels.

5) Have you tested expressions in the literature for $\eta_{NS} = \eta(\phi_{NS})$ to confirm that the gelation time in the small d_{NS} regime scales as $\eta_{NS}\phi_g^\alpha$?

Response Fig. 11. Relative background viscosity as a function of NS concentration. Data are from composite systems with small NS $d_{NS} = d$. Solid line is the Krieger–Dougherty form $(1 - \phi/\phi_m)^{-2.5\phi_m}$ [25], where we use $\phi_m = 0.58$.

As mentioned in the Discussion section, the gelation time of pure gels and binary composites (at small d_{NS} limit) can be respectively expressed as:

$$\begin{aligned} t_g^{\text{gel}} &\sim \eta_t \phi^\alpha, \\ t_g &\sim \eta_{NS} (\phi_g)^\alpha, \end{aligned} \tag{2}$$

where $\eta_{\text{NS}} = \eta_{\text{NS}}(\phi_{\text{NS}})$ refers to the viscosity of NS suspensions and $\eta_{\text{f}} = \eta_{\text{NS}}(0)$ to the fluid viscosity. In this way, the relative background viscosity η_{r} can be written in a normalized gelation time:

$$\eta_{\text{r}}(\phi_{\text{NS}}) \equiv \frac{\eta_{\text{NS}}(\phi_{\text{NS}})}{\eta_{\text{f}} = \eta_{\text{NS}}(0)} = \frac{t_{\text{g}}}{t_{\text{g}}^{\text{gel}}} \quad (3)$$

with $\phi = \phi_{\text{g}}$. For all composites with small $d_{\text{NS}} = d$, we plot η_{r} versus ϕ_{NS} in Response Fig. 11. Despite of various ϕ_{g} , we find that these data more or less collapse and roughly agree with suspension viscosity in the Krieger–Dougherty form $(1 - \phi/\phi_m)^{-2.5\phi_m}$ [25] with $\phi_m = 0.58$ (solid line). Note that the above argument does not consider the volume occupied by the colloids, as it is challenging to quantitatively exclude the gel part. Moreover, our simulations do not take hydrodynamics (e.g., stresslet and pair interaction) into account, so moderate deviation from theory is acceptable. We added relevant discussions to the manuscript (Page 7, line 407–410) and Supplementary Note 7.

6) Finally, I spotted a typo and suggest two formal corrections:

- p51322 “the counter line” → “the contour line”
- Maybe rather than “contour line”, “the iso- γ line” would be more understandable?
- In the caption of Fig4, I suggest to change “varies over time” to “time evolution” (twice).

Thanks and corrected.

-
- [1] Zhang, S. *et al.* Correlated rigidity percolation and colloidal gels. *Phys. Rev. Lett.* **123**, 058001 (2019).
- [2] Tsurusawa, H., Leocmach, M., Russo, J. & Tanaka, H. Direct link between mechanical stability in gels and percolation of isostatic particles. *Sci. Adv.* **5**, eaav6090 (2019).
- [3] Li, Y., Royer, J. R., Sun, J. & Ness, C. Impact of granular inclusions on the phase behavior of colloidal gels. *Soft Matter* – (2023).
- [4] Chubynsky, M. V. & Thorpe, M. F. Algorithms for three-dimensional rigidity analysis and a first-order percolation transition. *Phys. Rev. E* **76**, 041135 (2007).
- [5] Santos, A. P. *et al.* Granular packings with sliding, rolling, and twisting friction. *Phys. Rev. E* **102**, 032903 (2020). URL <https://link.aps.org/doi/10.1103/PhysRevE.102.032903>.
- [6] Liu, K., Kollmer, J. E., Daniels, K. E., Schwarz, J. M. & Henkes, S. Spongelike rigid structures in frictional granular packings. *Phys. Rev. Lett.* **126**, 088002 (2021).
- [7] Ferreiro-Córdova, C. *et al.* Stiffening colloidal gels by solid inclusions. *Soft Matter* **18**, 2842–2850 (2022).
- [8] Poon, W. & Haw, M. Mesoscopic structure formation in colloidal aggregation and gelation. *Adv. Colloid Interface Sci.* **73**, 71–126 (1997).
- [9] Cichocki, B. & Felderhof, B. U. Dynamic scattering function of a semidilute suspension of hard spheres. *J. Chem. Phys.* **98**, 8186–8193 (1993).
- [10] Koumakis, N. *et al.* Tuning colloidal gels by shear. *Soft Matter* **11**, 4640–4648 (2015).
- [11] Brady, J. F. The long-time self-diffusivity in concentrated colloidal dispersions. *J. Fluid Mech.* **272**, 109–134 (1994).
- [12] Sanchez, R. & Bartlett, P. Equilibrium cluster formation and gelation. *J. Condens. Matter Phys.* **17**, S3551 (2005).
- [13] Whitmer, J. K. & Luijten, E. Influence of hydrodynamics on cluster formation in colloid-polymer mixtures. *J. Phys. Chem. B* **115**, 7294–7300 (2011).
- [14] Fry, D., Mohammad, A., Chakrabarti, A. & Sorensen, C. M. Cluster shape anisotropy in irreversibly aggregating particulate systems. *Langmuir* **20**, 7871–7879 (2004).
- [15] Daneshfar, Z., Goharpey, F. & Foudazi, R. Depletion-induced interaction in concentrated bimodal suspensions of nanosilica in poly(ethylene glycol). *Rheol. Acta* **58**, 97–107 (2019).

- [16] Sikorski, M., Sandy, A. R. & Narayanan, S. Depletion-induced structure and dynamics in bimodal colloidal suspensions. *Phys. Rev. Lett.* **106**, 188301 (2011).
- [17] Chang, C. & Powell, R. L. Effect of particle size distributions on the rheology of concentrated bimodal suspensions. *J. Rheol.* **38**, 85–98 (1994).
- [18] Jamali, S., Yamanoi, M. & Maia, J. Bridging the gap between microstructure and macroscopic behavior of monodisperse and bimodal colloidal suspensions. *Soft Matter* **9**, 1506–1515 (2013).
- [19] Royall, C. P., Louis, A. A. & Tanaka, H. Measuring colloidal interactions with confocal microscopy. *J. Chem. Phys.* **127**, 044507 (2007).
- [20] Oversteegen, S. M. & Roth, R. General methods for free-volume theory. *J. Chem. Phys.* **122**, 214502 (2005).
- [21] Roth, R., Evans, R. & Dietrich, S. Depletion potential in hard-sphere mixtures: Theory and applications. *Phys. Rev. E* **62**, 5360–5377 (2000).
- [22] Gnan, N., Zaccarelli, E. & Sciortino, F. Casimir-like forces at the percolation transition. *Nat. Commun.* **5**, 3267 (2014).
- [23] Dagès, N. *et al.* Interpenetration of fractal clusters drives elasticity in colloidal gels formed upon flow cessation. *Soft Matter* **18**, 6645–6659 (2022).
- [24] Lekkerkerker, H. N. W., Poon, W. C.-K., Pusey, P. N., Stroobants, A. & Warren, P. B. Phase behaviour of colloid + polymer mixtures. *EPL* **20**, 559 (1992).
- [25] Krieger, I. M. & Dougherty, T. J. A mechanism for non-newtonian flow in suspensions of rigid spheres. *Trans. Soc. Rheol.* **3**, 137–152 (1959).

REVIEWERS' COMMENTS

Reviewer #2 (Remarks to the Author):

The authors answered all my questions and concerns in a concise and compelling manner. I thoroughly enjoyed the new manuscript, which will certainly resonate well with the community. I highly recommend its publication.

Reviewer #3 (Remarks to the Author):

The authors have adequately responded to my previous report. I can now recommend that their revised paper be accepted in its current form.

Reviewer #4 (Remarks to the Author):

I was appreciating the paper by Jiang and Seto at my first review and had only a few improvement suggestions. These comments have been taken into account, as well as (as far as I can tell from my expertise) the questions and suggestions of the other referees.

I did not detect any typo or misunderstanding from the newly added text or figures.